# Probiotics in Children with Asthma

**DOI:** 10.3390/children9070978

**Published:** 2022-06-29

**Authors:** Giorgio Ciprandi, Maria Angela Tosca

**Affiliations:** 1Allergy Clinic, Casa di Cura Villa Montallegro, 16146 Genoa, Italy; 2Allergy Center, Istituto Giannina Gaslini, 16146 Genoa, Italy; mariangelatosca@gaslini.org

**Keywords:** oral probiotics, asthma, wheezing, asthma exacerbation, add-on therapy

## Abstract

A type-2 immune response usually sustains wheezing and asthma in children. In addition, dysbiosis of digestive and respiratory tracts is detectable in patients with wheezing and asthma. Probiotics may rebalance immune response, repair dysbiosis, and mitigate airway inflammation. As a result, probiotics may prevent asthma and wheezing relapse. There is evidence that some probiotic strains may improve asthma outcomes in children. In this context, the PROPAM study provided evidence that two specific strains significantly prevented asthma exacerbations and wheezing episodes. Therefore, oral probiotics could be used as add-on asthma therapy in managing children with asthma, but the choice should be based on documented evidence.

## 1. Introduction

Asthma is an important issue for healthcare because of its high prevalence; consequently, asthma represents a relevant burden for the healthcare system [1]. In particular, asthma is the most frequent chronic disease in children and adolescents, affecting, for instance, ten percent of Italian school-aged subjects [2].

Asthma is a heterogeneous disease sustained by chronic inflammation of the respiratory tract, reversible airflow limitation, and bronchial hyperreactivity [3]. In addition, symptoms and bronchial obstruction may vary over time and in intensity.

Clinical, functional, and pathological features define asthma phenotypes and endotypes [3]. In this context, the type-2 phenotype is the most frequent in childhood; mostly, allergic asthma is predominant in this context [4].

Managing asthma, therefore, constitutes a remarkable commitment for pediatricians [5]. The asthma workup is based on demonstrating bronchial obstruction reversibility (or, if absent, bronchial hyperresponsiveness to stimuli), and measuring lung function, inflammation intensity, and symptom severity [6]. Asthma treatment has, as its goal, the achievement of asthma control by using relievers, such as symptomatic medications, mainly bronchodilator agents, and controllers, such as anti-inflammatory drugs, especially inhaled corticosteroids. In addition, biologics, such as blocking mediators of inflammation, are indicated for severe asthma [1,3].

However, diagnosis of asthma can be difficult in younger children, especially because of the difficulty of performing reliable spirometry, including any bronchodilator or bronchial stimulation test. As a result, the diagnosis of asthma is made on a clinical basis. In this context, the wheezing symptom is given due consideration.

Wheezing can be defined as a continuous whistling sound released by the airway during exhalation that results from a narrowing of caliber of the lower airway [7]. Infants suffering from wheezing constitute a diagnostic task for the pediatrician, also considering the high prevalence and lack tools available for objective testing in children with persistent wheezing [8]. The most common causes of wheezing in infants and preschoolers are asthma, gastric reflux, bronchiolitis, obstructive sleep apnea, and foreign body aspiration [9]. Less common causes are cardiac disease, pneumonia, and tracheobronchomalacia; rare causes are cystic fibrosis, vascular ring, immunodeficiency, ciliary dyskinesia, congenital bronchial atresia, and epiglottitis [10].

Wheezing in children five years or younger is a common problem. About 1/3 of European and American children aged 1–6 years present wheezing in the preceding six months, and almost half of children have at least one episode in the first six years of life [11]. Much effort has been expended to phenotype wheezing. Phenotyping considers the symptoms’ onset and duration. A common classification encompasses three main groups: transient early wheezing, non-atopic wheezing, and atopic wheezing/asthma [12]. In other words, such wheezing could be included in the diagnosis of asthma [1].

However, the resolution of bronchial obstruction remains crucial in treating an acute asthma attack using short-acting β2-adrenergic molecules. In addition, children with recurrent wheezing episodes usually benefit from daily low-dose inhaled corticosteroids [11].

It also must be noted that asthma attacks, including wheezing episodes, often follow respiratory infections [13]. In addition, allergic children frequently suffer from respiratory infections, as type-2 immune response is characterized by a defective defense against pathogens [14]. As a result, preventing respiratory infections represents a key aspect in the management strategy.

## 2. The Rationale for Probiotics in Pediatric Asthma

The prevalence of allergic diseases has shown a dramatic increase worldwide, so much so that the term allergy epidemics was coined [15]. Strachan proposed an intriguing hypothesis to explain this epidemiological problem in 1989 [16]. He associated a high prevalence of allergic diseases with the optimal hygiene status of affluent families. In contrast, people living in poor hygienic settings rarely were allergic, so he coined the term “hygiene hypothesis”. This hygiene hypothesis presupposed an imbalance of human microbiota. Gut microbiota represent the most relevant source for the development of the immune system in infants. Furthermore, a reduced antigenic “pressure” (as occurs in hygienic situation) promotes the maintenance of the type-2 response, characteristic of the fetus, to avoid maternal rejection. In this regard, it has been demonstrated that microbiota biodiversity may have an essential role in allergy rise [17]. Namely, a decline in biodiversity causes microbial deprivation, consequently disturbed immune response, and microbial imbalance. Biodiversity has been defined as “the variability among living organisms from all sources, including, among other things, terrestrial, marine, and other aquatic ecosystems and the ecological complexes of which they are part. It includes diversity within species, between species and ecosystems” [18].

Consequently, the Biodiversity Hypothesis assumes that contact with natural environments enriches the human microbiome, promotes immune balance, and protects from allergy and inflammatory disorders [18]. Consistently, asthmatic children may present dysbiosis of the intestinal and respiratory microbiome [19]. Dysbiosis may promote the start of inflammatory pathways and pitch in bronchial obstruction and airway hyperreactivity. However, exogenous factors can favorably modify the physiological airway microbiota composition (farming environment) or negatively (allergens, air pollutants), as demonstrated by several studies [20]. Moreover, intestinal dysbiosis significantly affects asthma pathogenesis [21]. Antibiotics, gastroprotective medications (mainly proton pump inhibitors), and other drugs may imbalance gut and lung microbiota. The consequent dysbiosis and reduced microbial diversity dysregulate the gut–lung axis, contributing to hypersensitivity and hyperreactivity against inhalant and food allergens. In this regard, the gut–lung axis is crucial in explaining the close relationship between enteric dysbiosis and allergic disorders. Therefore, it is enlightening to understand how dietary supplements may improve respiratory disorders [22]. These concepts are the basis to manipulate the immunity using natural remedies. Therefore, it has been envisaged that oral probiotics might restore the microbiota and immune balance.

The World Health Organization states that probiotics are defined as “live microorganisms which confer a beneficial effect on the host” [23]. Several studies evaluated probiotics in preventing allergic diseases and infections [24]. In addition, probiotics have been investigated as an add-on treatment in allergic disorders [25].

The benefits associated with probiotic supplementation depend on the multifaceted mechanisms of action. Maldonado Galeano and colleagues recently reviewed the beneficial effects of probiotics in humans [26]. After administration, oral probiotics interact with the intestinal epithelial cells (IECs) and immunocompetent cells through Toll-like receptors. This interaction stimulates the synthesis of mediators, cytokines, and chemokines. In particular, macrophage chemoattractant protein 1, released by IECs, induces the activation of the mucosal immunity, sustained by increasing immunoglobulin A-secreting cells of mucosal tissues. Probiotics also activate T cells. In addition, probiotics stimulate regulatory T cells to release IL-10, the main regulatory and anti-inflammatory cytokine [27]. Additionally, probiotics consolidate the intestinal barrier by increasing mucins, tight junction molecules, and goblet and Paneth cells. Finally, probiotics modulate gut microbiota by ensuring homeostasis and inhibiting the growth of pathogens [28].

In addition, the viability of probiotics is crucial to ensure their activity on innate immunity [29].

Therefore, probiotics could represent an exciting natural prevention and treatment strategy. They could expand type-1 response, downregulating IgE production, and reinforcing immune response to fight infections [24,25].

## 3. Systematic Reviews and Meta-Analyses of Probiotics in Asthma and Allergic Diseases

The hygiene hypothesis was based on the premise that a reduced pressure of “good” bacteria in the gut maintains the type-2 polarization in infants. The fetus grows in a type-2 environment to avoid maternal rejection, as it would be recognized as a non-self by the immune system. The external exposure, by diet and respiration, of microbes is the primary stimulus for maturing immunity; it occurs mainly in the gut [30]. As a result, several clinical trials investigated probiotics in allergy during the last decades. Different meta-analyses and systematic reviews evaluated the available studies.

Das and colleagues performed a systematic review of the additional use of probiotics in respiratory allergy [31]. These authors included 12 studies in the review, with 995 participants (547 for treatment and 488 as controls). The analysis revealed that probiotic supplementation significantly ameliorated the quality-of-life (QoL) scores in subjects suffering from allergic rhinitis. In addition, probiotic use prolonged the period free from asthma attacks and reduced the yearly rhinitis exacerbations. However, there was a high heterogeneity among the trials.

Lin and colleagues provided a systematic review and meta-analysis of probiotic supplementation in children with asthma [32]. This review selected 11 studies, including 910 children.

Children experiencing less asthma attacks belonged to the probiotics group (risk ratio 1.3, 95% confidence interval 1.06–1.59). Probiotic supplementation also reduced IL-4 (mean differences −2.34, 95% CI −3.38, −1.29) and increased interferon-γ (mean differences 2.5, 95% CI 1.23–3.76). However, the childhood asthma control test (cACT), asthmatic symptoms during the day and night, the number of symptom-free days, forced expiratory volume in the first second (FEV_1_) predicted, and peak expiratory flow (PEF) were unchanged after probiotic use. Therefore, the authors concluded that the outcomes could not recommend or advise against probiotic use in asthmatic children.

Du and colleagues conducted a meta-analysis of randomized controlled trials (RCT) to evaluate probiotic supplementary therapy for asthma, wheezing, and allergic rhinitis in children [33]. The study considered 17 RCTs, including 5264 children analyzed. The pooled data for the risk of developing asthma after probiotic supplementation showed no significant reduction compared with controls (risk ratio 0.86, 95% CI 0.73–1.01). However, *Lactobacillus rhamnosus* GG supplementation only reduced the occurrence of asthma (RR 0.75, 95% CI 0.57–0.99). The supplement in the postnatal group had a similar result.

Meanwhile, it failed to identify that probiotic supplementary therapy has a clear benefit to wheezing. Therefore, the authors concluded that the beneficial effects were based on the specific probiotic strain. It has to be underlined that this study privileges RCT concerning the prevention of asthma and wheezing.

Wei and coworkers conducted an RCT meta-analysis investigating the possible association between probiotics and incidence of asthma in infants [34]. Globally, 19 RCTs, enrolling 5157 children, fulfilled the inclusion criteria for the analysis. The results showed that probiotics did not affect the risk of asthma onset (RR 0.94, 95% CI 0.82–1.09) or wheezing (RR 0.97, 95% CI 0.88–1.06) compared with placebo. However, a subgroup analysis by asthma risk demonstrated that probiotics diminished wheezing incidence in allergic infants (RR 0.61, 95% CI 0.42–0.90). Therefore, the authors concluded that probiotic supplementation did not diminish the risk of asthma in infants. However, this study focused their attention on preventive studies alone.

Meirlaen and colleagues recently accomplished a narrative review regarding preventing and managing children with asthma and allergic rhinitis using probiotics, prebiotics, and synbiotics [35].

Unfortunately, the analysis of studies showed little evidence to recommend using this substance to prevent asthma and allergic rhinitis in children. However, this consideration derives from the high heterogeneity of the analyzed studies, the frequent low quality, and the use of a considerable quantity of different probiotic strains.

More recently, Chen and coworkers performed a meta-analysis of probiotics for respiratory allergies in children [36]. They included 15 randomized controlled trials including 1388 patients. The study demonstrated that probiotics enhanced quality of life and reduced symptom severity. However, the authors concluded that further research must thoroughly explain mechanisms of action.

The conclusions of these reviews should be carefully considered as the vital message relies on the concept that it is entirely incorrect to gather very different probiotics. It is as if we were analyzing antibiotics together: it makes no sense. Correctly, we have to consider each probiotic strain separately. Only reliable outcomes can be obtained when the analysis focuses on a single strain. Therefore, analyzing the data concerning a single probiotic strain is necessary. Each strain may exert specific activities depending on genetic, adaptative, immunological, and metabolic characteristics that make every strain unique, with potential specific efficacy and safety.

Considering this background, selecting probiotic strains with proven benefits for a specific disease is mandatory.

## 4. Probiotic Supplementation in Children with Asthma or Wheezing: The Most Relevant Studies

Giovannini and coworkers performed an RCT exploring the effects of a long-term (one year) consumption of fermented milk containing *Lactobacillus casei* in 187 preschoolers (2–5 years of age) with allergic asthma and/or rhinitis [37]. The findings showed no effect in asthmatic children. On the contrary, *Lactobacillus casei* supplementation diminished rhinitis episodes and shortened diarrhea episodes.

Rose and colleagues investigated the efficacy of *Lactobacillus rhamnosus* GG in preventing sensitization and asthma in 131 infants (6–24 months old) at risks, such as with at least two wheezing episodes and a first-degree family history of the atopic disease [38]. The supplementation lasted six months. The study design included a placebo arm. *Lactobacillus rhamnosus* GG significantly reduced the number of sensitizations both after six months and a further 6-month follow-up. In addition, subjects with pre-existing sensitization experienced more minor asthma complaints.

A Taiwanese RCT evaluated the clinical and immunological effects of an 8-week *Lactobacillus gasseri* A5 supplementation in 105 children (6–12 years old) with asthma and allergic rhinitis [39]. The probiotic use enhanced bronchial function, diminished asthma and rhinitis symptoms, and increased the ACT outcomes. Moreover, *Lactobacillus gasseri* A5 significantly diminished the production of TNF-α, IFN-γ, IL-12, and IL-13 by peripheral blood mononuclear cells.

An Italian double-blind placebo-controlled randomized study evaluated the effect of oral *Lactobacillus reuteri* administration on airway inflammation in 50 children (6–14 years old) with mild persistent asthma and house dust mite allergy [40]. The probiotic treatment lasted two months. *Lactobacillus reuteri* significantly decreased the fractional exhaled nitric oxide (FeNO) values and IL-2 levels; it significantly increased IL-10 levels. Thus, this RCT demonstrated that *Lactobacillus reuteri* dampened type-2 inflammation in asthmatic children.

The PANDA study evaluated the effects of a probiotic mixture containing *Bifidobacterium bifidum*, *Bifididobacterium lactis*, and *Lactococcus lactis*, administered during pregnancy and the first year of life [41]. It was an RCT. The probiotic supplementation reduced the incidence of eczema but not atopic dermatitis, such as eczema with sensitization. Therefore, a successive prospective, single-blind study was performed on 83 participants to investigate the outcomes at six years of age [42]. Unfortunately, the study did not show beneficial effects. This finding was in line with the results provided by the literature [43,44,45,46]. Therefore, pre- and perinatal probiotic supplementation seems to have scarce preventive effects. On the contrary, probiotic administration in asthmatic children appears more effective.

An Italian placebo-controlled, double-blind, randomized study consistently investigated the effects of a Bifidobacteria mixture containing *Bifidobacterium longum* BB536, *Bifidobacterium infantis* M-63, and *Bifidobacterium breve* M-16V in 40 children with seasonal allergic rhinitis and intermittent asthma [47]. The supplementation lasted four weeks. Children treated with probiotics significantly reduced respiratory symptoms and improved QoL, whereas the placebo group experienced worsened symptoms and QoL.

Ahanchian and colleagues evaluated a multi-strain synbiotic containing Lactobacillus casei, Lactobacillus rhamnosus, Streptococcus thermophilus, Bifidobacterium breve, Lactobacillus acidophilus, Bifidobacterium infantis, Lactobacillus bulgaricus, and fructooligosaccharides in 72 children (6–12 years old) with mild persistent asthma [48]. The study was randomized and placebo-controlled and lasted 60 days. Children in the synbiotic group had minor viral respiratory infections than the placebo group, but only during the first month. Consistently, actively-treated children used fewer bronchodilators.

A further Iranian RCT investigated the effects of a synbiotic compound containing *Bifidobacterium infantis* and fructooligosaccharide in school children (12 years of age or younger) with mild to moderate asthma [49]. The treatment lasted six months. The synbiotic supplementation induced a significant reduction in outpatient visits for asthma-related problems.

## 5. The PROPAM Study

Based on this background that globally supports probiotics as an add-on treatment in children with asthma, the PRObiotics in Pediatric Asthma Management (PROPAM) study investigated the impact of probiotics in the pediatric asthma management, considering the primary care setting [50].

Therefore, the PROPAM study was designed as a randomized, placebo-controlled, and double-blind trial to explore the efficacy of a probiotic mixture containing *Ligilactobacillus Salivarius* LS01 (DSM 22775) and *Bifidobacterium Breve* B632 (DSM 24706) in children with asthma or wheezing. The study evaluated their impact on asthma exacerbations and wheezing episodes the disease severity. The choice of these strains was based on the evidence concerning their efficacy and safety.

### 5.1. Bifidobacterium Breve B632: The Evidence

*Bifidobacteria* are commonly discovered in the intestinal microbiota of vaginally delivered and breastfeeding infants immediately after birth. It has been reported that infants with frequent colic have an intestine with a low quantity of *Bifidobacteria* and *Lactobacilli*. It is well known that these species exert significant anti-inflammatory and immune-stimulating effects [51]. The strain *Bifidobacterium breve* B632 possesses antimicrobial activity against gas-producing coliforms isolated from infants with colic [52]. The first in vivo step demonstrated a strain’s capability to colonize the intestinal microbiota. In this regard, a preliminary study on ten healthy children showed that both *Bifidobacterium breve* BR03 and *Bifidobacterium breve* B632 colonized the intestinal tract [53].

Therefore, a further study demonstrated that *Bifidobacterium breve* B632, added in cultures of colicky infants, inhibited the growth of *Enterobacteriaceae* [54]. Based on these findings, an in vivo study explored the efficacy of a probiotic mixture containing *Bifidobacterium breve* BR03 and *Bifidobacterium breve* B632 in preventing colic in 23 bottle-fed infants [55]. In addition, the probiotic supplementation significantly reduced the crying duration in the third month of treatment. This outcome underlines the importance of adequate treatment length, as probiotics are not symptomatic remedies but require sufficient time to carry out their beneficial effects. To confirm the effectiveness of these two strains, a very recent RCT study demonstrated that an 8-week *Bifidobacterium breve* BR03 and *Bifidobacterium breve* B632 supplementation improved insulin sensitivity in obese children and adolescents [56]. This study showed that changed gut microbiota could affect metabolic pathways.

### 5.2. Ligilactobacillus Salivarius LS01: The Evidence

An in vitro study showed that *Ligilactobacillus salivarius* LS01 (formerly *Lactobacillus salivarius*) reduced the release of type-2 cytokines, restoring the physiological type-1 immune response [57]. A first in vivo RCT study demonstrated that a 16-week supplementation with *Ligilactobacillus salivarius* LS01 in adult patients with atopic dermatitis significantly reduced the symptom severity assessed by the SCORAD [58]. In addition, the probiotic supplementation restored a physiological type-1/type-2 balance. The next step considered a high concentration of *Ligilactobacillus salivarius* LS01 (5 × 10^9^ CFU) and *S. thermophilus* ST10 (2 × 10^9^) associated with an innovative gelling complex in the 13 patients with atopic dermatitis [59]. Patients supplemented the probiotic for one month. The actively-treated patients experienced a significant reduction of symptom severity assessed by SCORAD. Consistently, a substantial decrease in fecal *S. aureus* was reported. A further study used a high concentration of this strain in the same disease model [60]. The outcomes confirmed previous findings.

Successively, *Lactobacillus salivarius* has been renamed *Ligilactobacillus salivarius* after thorough microbiological investigations [61].

Interestingly, it has been reported that *L. salivarius* LS01 supplementation reduced pro-inflammatory cytokine and, vice versa, increased anti-inflammatory cytokine production in patients with Parkinson’s disease [62].

Therefore, this probiotic strain has well-documented activity devoted to resolving inflammatory events and restoring microbiota balance.

### 5.3. Combined Bifidobacterium Breve B632 and Ligilactobacillus Salivarius LS01: The Evidence

The first clinical experience, conducted using these combined strains, was performed using the model of atopic dermatitis. An RCT included 48 patients with atopic dermatitis: 32 took the probiotic mixture for 12 weeks, and the remaining took a placebo [63]. Patients treated with probiotics experienced a significant reduction in SCORAD. In addition, probiotic supplementation reduced microbial translocation and restored the type-3/Treg cells ratio and type-1/type-2 balance [63].

A successive study investigated the model of chronic spontaneous urticaria [64]. An 8-week course of this probiotic mixture was administered to 38 patients, with spontaneous chronic urticaria recruited in two Italian allergy clinics. The results showed that this supplementation reduced symptom severity and improved QoL in about 30% of patients.

A clinical report concerning one patient with scalp rosacea demonstrated that this probiotic mixture associated with low-dose doxycycline, administered for eight weeks, significantly improved cutaneous and ocular symptoms [65]. This clinical improvement also persisted six months after discontinuation.

These positive outcomes were sustained by an in vitro study that demonstrated an immunomodulatory effect exerted by these two strains, alone or in combination [66]. Both singularly and combined strains reduced pro-inflammatory cytokines and increased the anti-inflammatory IL-10 produced by peripheral blood mononuclear cells recovered from patients with type-2 asthma. These outcomes are impressive, as they underline the capability provided by both strains to modulate the unbalanced immune response in allergic subjects. Namely, *Bifidobacterium breve* B632 and *Ligilactobacillus salivarius* LS01 exert a relevant antiallergic activity that doctors may favorably use to manage allergic patients.

## 6. Findings from the PROPAM Study

The study investigated as primary outcome the reduction in asthma exacerbations, assessing frequency and severity of asthma attacks [67]. In addition, the study investigated the reduction in wheezing episodes by a numerical measurement.

The study enrolled children aged between 6 to 14 years with asthma, diagnosed according to GINA criteria, and children aged between 3 and <6 years with recurring wheezing. Exclusion criteria were severe asthma, congenital or acquired immunodeficiency, cystic fibrosis, and chronic pulmonary diseases.

The study design was randomized, parallel-group, double-blind, and placebo-controlled. A sample size calculation was also performed.

The treatment phase lasted 16 weeks. For the first eight weeks, one sachet was administered in the morning and 1 in the evening; one sachet was administered per day for the last eight weeks.

The mixture contained *Lactobacillus salivarius LS01* (DSM 22775): 1 × 10^9^ CFU and *Bifidobacterium breve* B632 (DSM 24706): 1 × 10^9^ CFU, with maltodextrin and silicon dioxide used as bulking agents to yield a final weight of 2 g. The placebo contained 2 g of maltodextrin and silicon dioxide as bulking agents. The placebo powder was indistinguishable from the probiotic powder in appearance, taste, smell, and packaging.

Four hundred and forty-six children participated in the trial: 225 in the active group and 221 in the placebo group.

Globally, 50 (23.8%) children in the placebo group experienced at least one asthma exacerbation or wheezing episode, and 19 (9%) in the active group. In addition, 17 (8.1%) children in the placebo group and 5 (2.4%) in the active group had two acute respiratory episodes. There were 67 respiratory crises in the placebo group and 24 in the active group.

Children in the placebo group experienced a higher probability of having at least one asthmatic exacerbation or wheezing episode: (OR 3.17, 95% C.I. 1.8–5.6; *p* < 0.001). In addition, children in the placebo group had more probability of having two asthmatic exacerbations or wheezing episodes than children in the active group (OR 3.65, 95% C.I. 1.32–10.08; *p* = 0.013).

Consequently, the probiotic mixture diminished the odds of having an acute respiratory episode to a third. Furthermore, the probiotic mixture reduced nearly a quarter of the likelihood of having two crises.

Both treatments were well-tolerated, and no significant adverse events occurred.

## 7. The Relevance of PROPAM Study in Asthma Management

Primary care pediatricians commonly see asthma patients in daily practice [68]. In addition, exacerbations represent an important challenge in managing children with asthma [69].

Asthma exacerbation consists of an acute or sub-acute bronchial obstruction caused by chronic airway inflammation and bronchial hyperresponsiveness [70]. Bronchial obstruction and increased mucus production characterize asthma exacerbation. As a result, asthma symptoms, including wheeze, cough, dyspnea, and respiratory distress, worsen during asthma exacerbation [70]. In addition, the exacerbation outcomes usually include the need for systemic corticosteroids, urgent unscheduled care, emergency department or urgent care visits, and hospitalizations for asthma [71]. Therefore, asthma exacerbations require a thorough workup to identify factors associated. Furthermore, there is a close association between asthma exacerbation and asthma severity [72]. Additionally, wheezing is sustained by bronchial obstruction.

The leading cause of asthma attacks and wheezing episodes is an acute infection of the upper airway, mainly caused by a virus [73]. A viral infection induces the emergence of bronchial inflammation that triggers bronchial hyperreactivity and lower airway narrowing. Moreover, allergic subjects are likely to contract more frequent and severe infections than non-allergic subjects [73]. Consequently, a vicious circle is set up involving asthma, allergy, infections, and exacerbations [74]. These events result from type-2 immune response [75]. Therefore, it is important to take preventive measures aimed at restoring a physiological immune response [76].

Based on this background, there is a profound interest in therapeutic strategies that could manipulate immunity by restoring a physiologic type-1 response and dampening inflammation. Probiotics seem to respond to these requirements as there is documented evidence that they may rebalance immune responses and lower inflammatory events [77]. However, it is urgent to underline that the concept of probiotic benefit should be restricted only to the strains with documented proof of effectiveness. In this regard, RCT studies are welcome as they provide robust data that may support the use of probiotic strains in the management of pediatric asthma.

Another important aspect requires adequate attention: there is evidence that most probiotics fail the primary prevention of asthma [78,79]. Contrariwise, some convincing studies proved that probiotic supplementation reduced asthma severity in children with pre-existing asthma [47,48,49].

Therefore, the PROPAM study provided findings congruent with what was reported by previous RCT studies. Namely, the probiotic mixture containing *B. breve* B632 and *L. salivarius* LS01 significantly reduced the frequency of asthma attacks and wheezing episodes. Additionally, the probiotic mixture reduced the number of children with moderate/severe respiratory crises. However, the probiotic mixture did not affect the duration and treatment intensity of the maintenance therapy and needed therapies during respiratory crises.

Practically, children in the probiotic group experienced one-third of the acute respiratory episodes compared to control subjects. Moreover, the probiotic mixture significantly reduced the number of children with more than one respiratory attack. In addition, probiotics reduced the frequency of children experiencing more attacks to a quarter compared to placebo. Lastly, probiotics halved the number of children with moderate–severe attacks. These outcomes have relevant consequences in clinical practice. Asthma exacerbations, especially when complicated by hospitalization, implicate a relevant burden on the healthcare and have a negative impact on children and their families. In other words, such a dramatic reduction of respiratory crises provided by an RCT represents a relevant result. In addition, these findings were consistent with the positive effects of both strains, alone and in combination, in previous studies [54,55,56,57,58,59,60,61,62,63,64,65,66].

The immunologic effects provided by both strains may derive from the probiotic ability to produce tolerogenic peptides and improve epithelial barrier function [80,81]. *L salivarius* also upregulates intestinal barrier function [82]. In addition, Lactobacilli rebalance the type-1/type-2 ratio, restoring a physiological T helper 1 polarization [83,84]. Additionally, probiotics may expand the T regulatory immune response as a leading mechanism for suppressing type-2 responses to allergens [85].

The PROPAM study had some strengths, including randomization, the placebo control, a double-blind design, and the sample size calculation.

Additionally, it is important to highlight that the PROPAM study was conducted in a primary care setting. The obtained results accurately reflect what happens in daily practice. Therefore, the outcomes acquire a relevant practical significance.

In addition, as the age range was relatively wide, two post hoc analyses were performed to confirm the outcomes in preschoolers and schoolchildren [86,87]. Both analyses confirmed the results obtained in the whole population. Moreover, allergy was tested as an independent factor, as it is well known that allergic patients are particularly susceptible to infections [88]. Namely, the type-2 immune response reduces immune defenses against pathogens [89]. As a consequence, it is important to restore a physiological balance between types 1 and 2 [90]. Consistently, the post hoc analysis in the allergic population of the PROPAM study confirmed the efficacy in this subgroup [91].

Lastly, it has been very recently documented that the number of previous asthma exacerbations was the leading risk factor for another asthma exacerbation in preschoolers and schoolchildren [92,93]. Consequently, preventing an asthma exacerbation becomes a relevant target in asthma management. This outcome is significant in clinical practice [94].

On the other hand, the PROPAM study had some limitations, as declared in the manuscript. The limitations included the lack of mechanistic parameters and follow-up, and short treatment duration. As hypothesized by investigators, the lack of effectiveness on duration of exacerbations and care intensity could depend on the slow onset of action of probiotics. Probiotics may require a long period to modulate the immune system. Probably, the probiotic schedule should consider an adequate duration.

## 8. Discussion

It has been prospected that the increased prevalence of asthma and allergic diseases are promoted by reduced exposure to substances that promote immune system maturation [95]. In particular, the maturation of innate and acquired immunity occurs mainly at the gut level. In this regard, a number of bacteria, usually nonpathogenic, play an important effect for maturational purposes. All these concepts are included in the so-called Hygiene Hypothesis that has recently been revised and updated [96,97,98].

Thus, there would be the possibility to use “good bacteria” (probiotics) to manipulate the immune system.

Moreover, it has been proposed an intriguing hypothesis concerning the dual effect provided by low-molecular-weight bioregulators [99]. Namely, a murine model of ovalbumin-induced asthma demonstrated that bacterial cell wall fragments of glusaminylmuramyl dipeptide or lipolysaccharide exerted multidirectional effects, such as protective or harmful. This study emphasized the relevance of innate immunity in modulating allergic diseases and underscored the importance of the modality of exposure. Consistently, there are some findings supporting the possibility that prenatal supplementation of some probiotics could prevent allergies and asthma, mostly in children at high risk of allergy development [100]. However, the grade of evidence is still slight, and further rigorous studies need to provide convincing outcomes.

Regarding the add-on use of probiotics in children with actual asthma, the analysis of the existing literature on the use of probiotics in the management of pediatric asthma suggests that there may be a positioning of probiotics as add-on to conventional asthma therapy. Of course, it should be emphasized that probiotics cannot be considered drugs at all, and therefore cannot be counted in the pharmacological therapy of asthma. However, there are some findings that support the notion that probiotics may exert positive effects on the immune system and can reduce inflammatory events.

On the other hand, the studies that have been conducted are relatively few and have methodological biases. The duration of treatments, parameters evaluated, strains used, and clinical and functional characteristics of enrolled subjects varied across studies.

Thus, it is clear that more studies conducted according to a robust methodology are needed to reach more convincing conclusions about the real use of probiotics in pediatric asthma.

## 9. Conclusions

Currently, on the basis of the existing literature, it seems reasonable to assume that oral probiotics could be fruitful as an add-on treatment in children with asthma. Namely, specific probiotic strains can safely benefit children with asthma. However, the choice of probiotics should be based on documented evidence.

## Data Availability

Not applicable.

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
