# Peer review of "Probiotics in Children with Asthma"

_children, 2022, doi:10.3390/children9070978_

Round 1

Reviewer 1 Report

The article „Probiotics in children with asthma.” highlights the role of probiotics as a add-on treatment for asthma and other allergic diseases in children. The authors also highlighted possible recommendations in the introduction of two strains: Lactobacillus salivarius LS01 (DSM 22775) and 360 Bifidobacterium breve B632 (DSM 24706) for asthma children. I have read the paper with interest and feel that it is relevant for area of probiotics in allergic diseases.

I suggest few major revisions and comments are made below regarding the article.

  1. The authors should clarify the type of paper Article/Review because it is not clear.
  2. The type of “Font” should be according to the Template.
  3. Spelling and punctuation should be verified.
  4. There is no “Conclusion” in the end of the paper. Please add it.
  5. Conflict of interest declaration is at the end of the paper, please verify with the template
  6. Please add “Limitations of the PROPAM study” in discussion and add a “Discussion” part in the paper. Please revise.
  7. References should be modified according to the journal requests for publication.

Author Response

Firstly, we would thank the Reviewer for the helpful comments.

The article „Probiotics in children with asthma.” highlights the role of probiotics as a add-on treatment for asthma and other allergic diseases in children. The authors also highlighted possible recommendations in the introduction of two strains: Lactobacillus salivarius LS01 (DSM 22775) and 360 Bifidobacterium breve B632 (DSM 24706) for asthma children. I have read the paper with interest and feel that it is relevant for area of probiotics in allergic diseases.

I suggest few major revisions and comments are made below regarding the article.

  1. The authors should clarify the type of paper Article/Review because it is not clear.

R Many thanks for this comment. The paper is a review of the topic.

  1. The type of “Font” should be according to the Template.

R We adjusted it.

  1. Spelling and punctuation should be verified.

R We checked the text

  1. There is no “Conclusion” in the end of the paper. Please add it.

R Many thanks for this comment. We added the Conclusion section.

  1. Conflict of interest declaration is at the end of the paper, please verify with the template

R We change accordingly.

  1. Please add “Limitations of the PROPAM study” in discussion and add a “Discussion” part in the paper. Please revise.

R Many thanks for this comment. We implemented the manuscript with these parts.

  1. References should be modified according to the journal requests for publication.

R We modified references as suggested.

Reviewer 2 Report

The article is deal with the probiotics in children with asthma. The topic discussed is very important for the treatment and prevention of asthma.

I would like to make a few comments:

1)      line 52 and further in the text: Preschool wheezing

             It is better to use another phrase (Wheezing of children five years or younger or any other)

2)      line 53: have wheezed

            It is better to use “have wheezing”

3)      lines 79-90: When you talk over the “hygiene hypothesis” cite the recent publication and  discuss the idea  that bacterial fragments may be useful in prevention asthma and may worsen asthma during an acute course:

Guryanova, S.V.; Gigani, O.B.; Gudima, G.O.; Kataeva, A.M.; Kolesnikova, N.V. Dual Effect of Low Molecular Weight Bioregulators of Bacterial Origin in Experimental Model of Asthma. Life 2022, 12, 192. https://doi.org/10.3390/life12020192

4)      When you are writing about prevention of respiratory infections as an essential role in the therapeutical strategy cite and discuss the recent publication:

Colquitt AS, Miles EA, Calder PC. Do Probiotics in Pregnancy Reduce Allergies and Asthma in Infancy and Childhood? A Systematic Review. Nutrients. 2022 Apr 28;14(9):1852. doi: 10.3390/nu14091852

Author Response

Firstly, we would thank the Reviewer for the helpful comments.

The article is deal with the probiotics in children with asthma. The topic discussed is very important for the treatment and prevention of asthma.

I would like to make a few comments:

1)      line 52 and further in the text: Preschool wheezing

             It is better to use another phrase (Wheezing of children five years or younger or any other)

R Many thanks for this comment. We changed the term.

2)      line 53: have wheezed

            It is better to use “have wheezing”

R we changed the sentence.

3)      lines 79-90: When you talk over the “hygiene hypothesis” cite the recent publication and  discuss the idea  that bacterial fragments may be useful in prevention asthma and may worsen asthma during an acute course:

Guryanova, S.V.; Gigani, O.B.; Gudima, G.O.; Kataeva, A.M.; Kolesnikova, N.V. Dual Effect of Low Molecular Weight Bioregulators of Bacterial Origin in Experimental Model of Asthma. Life 2022, 12, 192. https://doi.org/10.3390/life12020192

R Many thanks for this comment. We considered the suggested article and discussed this point.

4)      When you are writing about prevention of respiratory infections as an essential role in the therapeutical strategy cite and discuss the recent publication: 

Colquitt AS, Miles EA, Calder PC. Do Probiotics in Pregnancy Reduce Allergies and Asthma in Infancy and Childhood? A Systematic Review. Nutrients. 2022 Apr 28;14(9):1852. doi: 10.3390/nu14091852

R Many thanks for this comment. We cited and discussed the suggested paper. However, the suggested paper considered the use of probiotics in the primary prevention of asthma, whereas our aim concerned the secondary prevention in children with actual asthma.

Round 2

Reviewer 1 Report

The article „Probiotics in children with asthma.” highlights the role of probiotics as a add-on treatment for asthma and other allergic diseases in children. The authors also highlighted possible recommendations in the introduction of two strains: Lactobacillus salivarius LS01 (DSM 22775) and 360 Bifidobacterium breve B632 (DSM 24706) for asthma children. I have read the paper with interest and feel that it is relevant for area of probiotics in allergic diseases.

I suggest few minor revisions regarding English style and formating.
